# Functional Network Mapping Reveals State-Dependent Response to IGF1 Treatment in Rett Syndrome

**DOI:** 10.3390/brainsci10080515

**Published:** 2020-08-03

**Authors:** Conor Keogh, Giorgio Pini, Ilaria Gemo, Walter E. Kaufmann, Daniela Tropea

**Affiliations:** 1Department of Surgery, University Hospital Limerick, V94 F858 Limerick, Ireland; keoghco@tcd.ie; 2Tuscany Rett Center, Ospedale Versilia, SS 1 Via Aurelia, 335, 55041 Camaiore LU, Italy; pinivg@gmail.com (G.P.); ilaria.gemo@uslnordovest.toscana.it (I.G.); 3Department of Human Genetics, Emory University School of Medicine; Whitehead Biomedical Research Building, 615 Michael Street, Suite 301, Atlanta, GA 30322, USA; walter.e.kaufmann@emory.edu; 4Neuropsychiatric Genetics, Trinity Centre for Health Sciences, St. James Hospital, D08 NHY1 Dublin, Ireland

**Keywords:** Rett Syndrome, IGF1, EEG, network, electrophysiology, machine learning

## Abstract

Rett Syndrome (RTT) is a neurodevelopmental disorder associated with mutations in the gene *MeCP2*, which is involved in the development and function of cortical networks. The clinical presentation of RTT is generally severe and includes developmental regression and marked neurologic impairment. Insulin-Like growth factor 1 (IGF1) ameliorates RTT-relevant phenotypes in animal models and improves some clinical manifestations in early human trials. However, it remains unclear whether IGF1 treatment has an impact on cortical electrophysiology in line with *MeCP2*’s role in network formation, and whether these electrophysiological changes are related to clinical response. We performed clinical assessments and resting-state electroencephalogram (EEG) recordings in eighteen patients with classic RTT, nine of whom were treated with IGF1. Among the treated patients, we distinguished those who showed improvements after treatment (responders) from those who did not show any changes (nonresponders). Clinical assessments were carried out for all individuals with RTT at baseline and 12 months after treatment. Network measures were derived using statistical modelling techniques based on interelectrode coherence measures. We found significant interaction between treatment groups and timepoints, indicating an effect of IGF1 on network measures. We also found a significant effect of responder status and timepoint, indicating that these changes in network measures are associated with clinical response to treatment. Further, we found baseline variability in network characteristics, and a machine learning model using these measures applied to pretreatment data predicted treatment response with 100% accuracy (100% sensitivity and 100% specificity) in this small patient group. These results highlight the importance of network pathology in RTT, as well as providing preliminary evidence for the potential of network measures as tools for the characterisation of disease subtypes and as biomarkers for clinical trials.

## 1. Introduction

### 1.1. Rett Syndrome

Rett syndrome (RTT) is a neurodevelopmental disorder characterised by initial normal development followed by spoken language and fine motor regression, gait impairment and hand stereotypic movements [1,2,3]. Two major presentations are recognised: classic/typical and variant/atypical. RTT is the second most common cause of intellectual disability in females, affecting 1 in 10,000–20,000 live female births [4]. RTT results in severe neurobehavioral impairment and associated clinical features throughout life. These include epilepsy, scoliosis and breathing abnormalities [5,6]. There are currently no disease-modifying therapies, and management is mainly symptomatic [7,8]. The majority of patients carry mutations in the *MECP2* gene, which encodes methyl CpG-binding protein 2, a chromatin binder and transcription regulator [9,10]. Rarer variants are associated with mutations in *CDKL5* (Cyclin-Dependent Kinase-Like 5), a regulator of MeCP2, and *FOXG1* [11,12]. Dysfunction of the *MECP2* gene results in impaired brain development, neuronal structure and synaptic function [13], with consequent abnormalities in network function [14].

### 1.2. Network Pathology

There is increasing recognition of the role of brain network pathology in human disease [15]. Rather than localised disturbances, symptoms are linked to abnormal interactions across networks of neural tissue [16]. Evaluation of the cortical network architecture using signal processing techniques opens the possibility of better characterisation of the network-level pathology underlying many neurologic and psychiatric disorders [17]. Further, network-based approaches have demonstrated promise in the development of diagnostic [18] and prognostic [19] biomarkers of neuropsychiatric disease. The potential utility of EEG-based measures in RTT and related disorders has recently gained attention [20], including evidence of abnormal spectral power profiles relative to typically developing controls [21] and deterioration in measures over time [22]. It has been demonstrated that RTT subtypes are associated with specific network profiles [23], suggesting that network-based approaches may facilitate the identification of further biomarkers for disease characterization and for improved design of intervention studies.

### 1.3. IGF1 Treatment

In an effort to develop treatments for RTT, molecules related to *MeCP2* have been targeted therapeutically. In preclinical studies, neurotrophic factors such as brain-derived neurotrophic factor (BDNF) were shown to counteract the effects of *MeCP2* dysfunction on RTT-relevant phenotypes, suggesting that approaches targeting this pathway may be promising [24,25,26]. BDNF is not therapeutically useful due to its pharmacokinetics [27], but other factors such as insulin-like growth factor (IGF1), which can be delivered therapeutically and cross the blood-brain barrier [28], have shown an ability to improve symptoms in early RTT trials and relevant studies in animal models [29,30,31,32,33,34,35].

These results have led to multiple clinical trials involving IGF1 and analogues, such as trofinetide [36] and recombinant IGF1. Safety trials have shown no effects on seizures, as patients treated before with anti-epileptic drugs (AED) tolerated the treatment, and it was not necessary to adjust the dosage [37]. Trials of IGF1 have shown conflicting results. An early trial demonstrated safety and clinical efficacy [37,38]. A follow-up phase 2 trial did not show clear benefit in the primary outcome measures; however, there were improvements in some secondary endpoints related to behaviour and communication, while some other measures worsened, and spectral power on EEG was shown to change over time [22]. Notably, these trials showed marked heterogeneity in treatment response, despite adequate subject matching at baseline. This response variability suggests that there may be a subpopulation of treatment-susceptible individuals which is not identifiable by routine clinical assessments.

### 1.4. Targeting Networks in Rett Syndrome

Considering that IGF1 can be beneficial to some individuals with RTT, and that IGF1 can modulate the maturation of cortical circuitry [39] and dendritic spine plasticity [35], we hypothesised that treatment with IGF1 would impact network profiles in RTT, and that there may be a relationship between network activity and clinical response. Further, we hypothesised that the characterisation of the network architecture at baseline in individuals with RTT would make it possible to identify treatment-susceptible patients and to predict treatment response.

To test our hypotheses, we used noninvasive EEG recording to investigate network features and their association with clinical response to IGF1. We found an association between IGF1 administration and specific changes in network architecture, and between network activity and clinical response to treatment. Based on the results obtained in this small group, we conclude that analyses of specific network activity parameters may identify treatment-responsive RTT subpopulations which would not be evident from clinical assessments. These findings have implications for the development of new treatments for RTT if borne out in larger cohorts.

## 2. Materials and Methods

### 2.1. Subjects

Patients were recruited from the Tuscany Rett Centre, Viareggio, Italy. All studies were conducted in accordance with the Declaration of Helsinki and approved by the Tuscany Child Ethical Committee (Approval ID: 12720). Patients’ families consented to data collection and use for scientific purposes.

Ten individuals were recruited for a trial with recombinant human IGF1, as described by Pini et al. 2016 [37]. One participant was excluded from the present analysis due to an incomplete electrophysiological dataset, so nine patients were analysed in this study. All participants had a classic phenotype and a verified pathogenic *MECP2* mutation. Matched individuals with RTT were recruited as controls. These patients did not receive any investigational treatment, i.e., neither drug nor placebo. All participants were subjected to the same schedule of assessments and continued to receive standard treatment.

### 2.2. IGF1 Administration

Subcutaneous human recombinant IGF1 (mecasermin) was administered twice daily for 20–24 weeks. Six patients described in this study were treated with IGF1 for 24 weeks. Four additional patients were treated for 20 weeks [37]. There was no correlation between the duration of the treatment and outcomes. For all patients, a dose of 0.05 mg/kg was administered during the first and last weeks, and a dose of 0.1 mg/kg during the intervening weeks. This dose is in line with that recommended for IGF1’s endocrine indications, and safety at this dosing level has been previously demonstrated [38]. For one of the patients treated with IGF1, the EEG recording were not suitable for the present study; hence, the study is based on data from nine patients.

### 2.3. Data Collection

Subjects were evaluated clinically and from EEGs performed at baseline (pretreatment) and at 12 months after treatment initiation.

#### 2.3.1. Clinical Characterisation

Disease severity was assessed clinically using the International Scoring System (ISS) [40,41] at baseline and at 12 months. Clinical assessments were performed by two independent assessors and averaged for each patient. Clinical assessment scores were used to match IGF1 treated and untreated groups at baseline and for evaluating changes following treatment.

The treated group was subdivided into responders or nonresponders on the basis of changes in severity score from baseline to end of treatment (Table 1). Treatment response was defined as any reduction in ISS scores from baseline to 12 months, as determined by two independent assessors.

The components of the ISS severity score are outlined in Appendix B.

#### 2.3.2. Electrophysiological Recordings

Electroencephalographic (EEG) data were recorded using an eight-electrode montage. Recordings were made for a minimum of 20 min on awake subjects, at rest in a dedicated recording environment with minimal sensory stimulation, following a period of adaptation to the environment. EEG data were preprocessed in order to ensure accurate feature extraction and to eliminate artefacts, including those created by motion and epileptiform activity. Details on electrode montage and data processing are given in Appendix C.

### 2.4. EEG Feature Extraction

Profiles of electrophysiological features were derived from recorded EEG data using custom MatLab scripts as described in Keogh et al. 2018 [23]. All scripts used in this analysis are available at https://github.com/conorkeogh/IGF_Response/.

#### 2.4.1. Spectral Power

Power spectra were calculated using Fourier Transforms of preprocessed EEG data. Overall power, measured in decibels, was assessed using the mean of individual channel spectra. Individual frequency bands were assessed by isolating these bands and calculating relative power normalised to overall power [42].

#### 2.4.2. Profiles

Distribution of activity was assessed by evaluating the relative activity at each hemisphere. This was calculated by subtracting spectral power measures in the right hemisphere from those in the left. Detailed profiles of power distribution were derived by analysing power across the whole hemisphere, at each individual electrode and within frequency bands.

#### 2.4.3. Network Measures

Network function was analysed by evaluating interactions between electrodes [43]. Measures of interelectrode coherence were derived for each electrode pair through calculation of the cross-spectrum of the channels normalised by the power spectra of both channels, repeated for each unique pair:(1)$$C\left(\omega \right)=\frac{S_{\mathrm{xy}}{\left(\omega \right)}^{2}}{S_{xx}\left(\omega \right)S_{\mathrm{yy}}\left(\omega \right)}$$

Dimensionality reduction of connectivity measures was performed via principal component analysis [44]. Network architectures were assessed and compared between groups by examining the covariance of interelectrode coherence measures and associated loadings of network measures (i.e., the principal components of the coherence measures) [45]. This allowed the overall architecture of cortical networks to be compared between groups while limiting the number of statistical tests required. Pairwise comparisons of individual electrode pairs were performed on a post hoc, exploratory basis to better characterise the nature of any differences seen in comparisons of principal components.

Patient recruitment and clinical evaluations were performed by personnel who were not involved in EEG analysis. Processing and feature extraction algorithms were applied to all recordings without consideration of clinical features, disease severity, treatment status, genetics or any other clinical parameter.

All data processing was performed blinded to patient characteristics.

### 2.5. Statistical Analyses

The effect of IGF1 treatment was investigated using a two-way, mixed-factor, repeated-measure ANOVA to assess whether an interaction between treatment group and timepoint existed. Nonparametric statistical tests were used in subsequent post hoc analyses to avoid assumptions of normality. Mann-Whitney U tests were used for pairwise comparisons. Wilcoxon signed-rank tests were employed for paired data in within-group analyses. All tests were two-tailed.

All statistics are reported as mean +/− standard deviation.

#### 2.5.1. Effect of Treatment

Treated (*n* = 9) and untreated *n* = 9) groups were compared. Treatment status (treated vs. untreated) was used as a between-group measure, with timepoint (baseline vs. twelve months) as a within-subjects factor. The treated group was then subdivided into responders (*n* = 5) and nonresponders (*n* = 4), and these subgroups compared in order to assess whether electrophysiological measures varied with clinical response.

#### 2.5.2. Predicting Treatment Response

Machine learning techniques were used to assess the predictive ability of EEG parameters identified in those who responded to IGF1 treatment.

#### 2.5.3. Model Training

A support vector machine classifier was trained on all network features identified as differing to a statistically significant degree between responders and nonresponders before treatment on post hoc analysis. These features were used to define the planes of a potential biomarker feature space [46]. A training algorithm identified the plane within this space that best separated the two groups. This was then used to classify further data.

#### 2.5.4. Validation

A five-fold cross validation approach was used to validate the model and to assess its performance when presented with a new set of data, as in a clinical scenario [47]. The data for the treated group was randomly partitioned into five subsets (i.e., two subjects were randomly excluded from model training and used for testing at each iteration). A classifier was trained on four subsets, and the fifth, not involved in model training, was used to assess the classifier’s prediction accuracy. This was then repeated, with a model trained on each set of four subsets and tested on the remaining subset, and the overall accuracy of the modelling approach was tested.

#### 2.5.5. Sequential Elimination

Model training and validation was repeated, eliminating the network feature with the least statistically significant difference from the model at each repetition; the resulting accuracies were then assessed. This allowed us to identify the features driving the predictive ability of the machine learning classifier. This could then be used to determine the minimum dataset required to accurately predict treatment response, and to provide insight into the mechanisms underlying this response.

## 3. Results

### 3.1. Demographics

The overall group had a median age of 6 years (interquartile range 6), with no statistically significant difference between subgroups (treated: 4, IQR 5; untreated: 8, IQR 7; *p* = 0.35, Mann-Whitney U test). The overall group had a mean clinical severity score (International Scoring System, ISS [37,38]—see Appendix A) of 17.4 +/− 4.5, with no statistically significant difference between subgroups (treated: 17.3 +/− 3.7; untreated: 17.5 +/− 5.4; *p* = 0.95, Mann-Whitney U test). Individual-level demographic and clinical profiles are shown in Table 1, and ISS scores over time are shown in Figure 1. Notably, earlier treatment did not appear to be associated with treatment response in this small group; however, there appeared to be a trend towards increased resistance with higher baseline ISS; this suggests that treatment response and resistance may be a function of the state of the underlying network, rather than simply dependent on the age at which treatment is started.

### 3.2. Treatment Effect

There were no differences between treated and untreated groups in terms of overall power spectrum, distribution of power across the scalp and pattern of hemispheric asymmetry at baseline or at 12 months (full pairwise comparisons shown in Appendix A). There were also no statistically significant differences within treated or untreated groups in these measures between timepoints (Figure 2). There were no statistically significant differences in asymmetry measures between or within groups. Notably, these measures demonstrated a large degree of interindividual variability; the plots suggest a trend towards differences in power spectra between groups, as well as potential differences in the distribution of activity, as reflected by asymmetry. While these results were not statistically significant, they may reflect coarser changes relative to network-level assessments.

Analyses of network loadings, using a two-way repeated measures ANOVA with treatment status as a between-groups factor and timepoint (baseline vs. 12 months) as a within-subjects factor, demonstrated a significant interaction of treatment status and timepoint (*p* < 0.0001), suggesting an effect following treatment. The effect of time alone was not statistically significant (*p* = 0.051).

Given the apparent time-dependent effect of treatment, treated and untreated groups were compared at 12 months to investigate the nature of the effect of IGF1 treatment on network measures. To further delineate the effects of treatment, the treated and untreated groups were analysed using a within-group comparison, comparing baseline and 12-month timepoints.

A comparison of network loadings between treated and untreated subjects at twelve months showed a statistically significant difference between groups (*p* < 0.0001, Mann-Whitney U test). However, pairwise comparison of electrode pairs (Appendix A) showed no significant differences, suggesting that this difference was not driven by localised changes at single electrode pairs. Further, the analysis of network measures revealed a statistically significant difference in network architecture within the treated group following treatment (*p* < 0.0001, Wilcoxon signed rank test), supporting a treatment-induced alteration in network measures. Post hoc analyses of individual electrode again pairs found no differences in the coherence of individual pairings in the overall power spectrum (results of individual comparisons are provided in Appendix A). These data suggest that the observed difference in overall architecture was not driven by specific electrode pairs, but rather, that it reflected a global difference in connectivity patterns.

Consistent with an effect driven by an interaction between treatment group and timepoint, a within-group comparison of network loadings between timepoints in those who did not receive treatment showed no significant difference in network measures. Likewise, there were no significant differences at the level of individual electrode pairings or frequency bands (results of individual comparisons are included in Appendix A). These results indicate that there were no alterations in any network measures in those who did not receive IGF1 treatment.

Notably, there was a statistically significant effect of treatment group independent of timepoint (*p* < 0.0001), suggesting a difference in network architecture between treatment groups at baseline, and indicating that electrophysiological differences between these groups were not anticipated by the clinical evaluation. These network-level differences at baseline were investigated in post hoc analyses by evaluating coherence measures within the overall power spectrum at the level of individual electrode pairings (Appendix A). We found statistically significant differences in specific regions, indicating that that the primary drivers of the overall difference in network connectivity were temporal and occipital-frontal connections (threshold at *p* < 0.05, Mann-Whitney U test). A pairwise comparison of network measures across all electrode pairs is shown in Appendix A, and the spatial distribution of the electrode pairs driving network differences is shown in Appendix A. Further analyses of connectivity patterns within individual EEG bands supported differences in occipital and frontal networks within individual frequency bands (Appendix A).

The network architecture, visualised as covariance between connectivity measures at electrode pairs in the overall power spectrum, is shown in Figure 3. High resolution covariance matrices of network interactions across the overall spectrum and individual power bands are shown in Appendix A.

### 3.3. Responder Status

Within the treated group, there were no statistically significant differences in overall power spectra, power distribution or asymmetry measures between responders and nonresponders prior to treatment or at twelve months (Figure 4; full pairwise comparisons shown in Appendix A). There were also no statistically significant differences in these measures within groups between timepoints.

The analysis of network loadings using a two-way, repeated measures ANOVA with treatment response (responder vs. nonresponder) as a between-groups factor and timepoint (baseline vs. 12 months) as a within-subjects factor demonstrated a significant interaction of treatment response and timepoint (*p* < 0.0001), suggesting a difference in electrophysiology following treatment in those that showed a clinical response. The effect of time alone was not statistically significant (*p* = 0.081). 

A post hoc analysis of network loadings showed a statistically significant difference between responders and nonresponders following treatment (*p* < 0.0001, Mann-Whitney U test), consistent with a significant interaction of responder status and timepoint on network architecture. An evaluation of individual electrode pairs demonstrated no significant differences in coherence in the overall power spectrum between responder and nonresponder groups (Appendix A), suggesting that this difference in network architecture was not driven by major differences between specific electrode pairs, but rather, was dependent on a global effect.

However, there was a statistically significant effect of treatment response independent of timepoint (*p* = 0.002), suggesting a difference in network architecture between responder groups at baseline, indicating that electrophysiological differences between these groups were not anticipated by clinical evaluation. The network architecture, visualised as covariance between connectivity measures at electrode pairs in the overall power spectrum, is shown in Figure 5. High resolution covariance matrices of network interactions across the overall spectrum and individual power bands are shown in Appendix A.

Network differences between responder groups at baseline were explored using post hoc analyses to investigate whether network measures may have utility in predicting treatment response. The network differences were primarily driven by occipital interhemispheric and frontal-occipital connections (Appendix A). The locations of electrode pairings found to have a statistically significant difference in coherence in the overall power spectrum between groups (threshold at *p* < 0.05, Mann-Whitney U test) are shown in Appendix A.

### 3.4. Predicting Response

A support vector machine was trained using pretreatment measures of occipital interhemispheric and long-range frontal-occipital connectivity (Appendix A), as identified in post hoc analysis of baseline differences in those that responded clinically to treatment. We assessed the ability of this model to predict the treatment response from pretreatment network connectivity measures using five-fold cross-validation.

The model correctly predicted treatment response with 100% accuracy (sensitivity: 100%; specificity: 100%). A confusion matrix demonstrating prediction accuracy is shown in Figure 6A. The distributions of the connectivity measures involved in the model are shown in Figure 6B. Notably, the predictive accuracy appeared to be predominantly driven by occipital interhemispheric coherence. A model trained on occipital interhemispheric coherence alone predicted treatment response with 100% accuracy. The small sample sizes used are likely to have resulted in some overfitting, though the performance seen is indicative of the potential value of EEG biomarkers in targeting novel treatments.

## 4. Discussion

In this study, we used signal analyses of noninvasive resting state EEGs to characterise cortical network profiles in patients with Rett Syndrome treated with IGF1 and untreated patients (controls). We found that IGF1 treatment was associated with selective changes in network parameters, and that these changes were related with clinical response, suggesting that the clinical effects of IGF1 may be mediated by alterations in network dynamics. Furthermore, we showed that response to IGF1 appeared to be dependent on the state of cortical networks prior to treatment, and that analysis of network measures at baseline could accurately predict treatment responsiveness. These findings in a relatively small group of patients with RTT support network analysis as a valuable avenue for the development of more targeted treatment approaches.

Our results also revealed an unanticipated level of variability in network parameters within clinically-matched treatment groups at baseline, indicating that electrophysiological subpopulations may exist within these groups without clinically evident differentiation. Given the apparent state-dependent effects of IGF1 treatment, the existence of this baseline network variability underscores the need for greater characterisation and standardisation of electrophysiological phenotyping methods in order to ensure that novel treatments target subpopulations that are most likely to benefit from them.

### 4.1. IGF1 Treatment

The effects of IGF1 treatment have demonstrated variability in different clinical studies. No significant safety issues have been reported, however. This variability may be attributed to a number of factors, including the timeline of administration, the diverse presentations of patients with RTT, the limited number of patients enrolled in the studies and the measures of functional outcomes considered.

In particular, IGF1 administration times deserve additional attention. The dosage and duration of the treatment in the studies on patients with RTT were selected on the basis of the regimen of IGF1 treatment for dwarfism. Recent studies in cells derived from patients with autism [48] and in a clinical case with RTT treated with two cycles of IGF1 [49] suggested that prolonged treatment with IGF1 may not be ideal to produce positive benefits, at least as measured in terms of severity scores and clinical evaluation. Considering that there were no differences between the efficacy reported in patients treated for 20 or 24 weeks, and that the majority of the benefits were recorded within three months from the start of the treatment [49], it is likely that shorter administration (three months) may be more effective. More quantitative and objective measures are necessary to assess the effects of IGF1 treatment over time to further explore this issue.

### 4.2. Treatment-Induced Network Effects

Our results indicate that treatment with IGF1 leads to alterations in measures of cortical network function. The strong interaction between treatment status and time-point indicates a significant effect of IGF1 treatment on the overall network architecture of patients with RTT. Interestingly, this network difference appears to be a global phenomenon, as analyses of differences at the level of individual electrode pairs demonstrated that the treatment effect was not driven by significant differences at individual electrodes. The impact of IGF1 on network architecture was further supported by evidence of network differences in individuals who showed a clinical response to treatment (Responders), supporting a relationship between network changes and clinical outcomes. Further, consistent with previous evidence that network profiles are stable over time in RTT, the untreated group showed no evidence of network changes over the 12-month period, supporting the idea that the changes seen in those that received IGF1 is a true treatment effect.

Although it is difficult to draw causal relationships, mediation of IGF1′s effects via network activity is consistent with its role at different levels of synaptic activity [50]. Our data suggest that the clinical effects of IGF1 may be mediated by its action on dysfunctional networks because of MeCP2 deficits.

Recognition of the potential for EEG measures as biomarkers in RTT has grown recently [20] in light of the underlying disease physiology. Notably, a recent trial of IGF1 demonstrated alterations in spectral power measures within specific EEG bands at specific locations [22], particularly in frontal regions. While these findings support the notion that IGF1 mediates its effects through modulation of network parameters, we did not find overall differences in spectral power parameters following treatment. When examining individual EEG bands at individual electrode locations, there were some differences noted in frontoparietal regions, though no such differences were evident on the overall spectrum, and none survived correction for multiple comparisons. Further, there was no evidence for any statistically significant differences in interhemispheric asymmetry, i.e., a measure of the distribution of activity within the brain which has been associated with RTT [22], in our analyses. All of these pairwise results are available in the Appendix A.

These apparent differences in electrophysiological outcomes may be attributed to the low sample size; the suggestion of a frontoparietal signal may have translated into a stronger result in a larger group. This suggests that the network-level metrics may offer a more reliable means of differentiating RTT patients than coarser measures of spectral power in individual regions. Further, differences in the context of acquisition may be relevant; differences in spectral power were evident while watching a video [22], but were not apparent here, where there was no visual stimulus. Similarly, factors such as the status of the patients during the acquisition of the EEG signal must be taken into account. In the present study, many patients were seen multiple times per year and interacted with the personnel for a long time before the recordings. This procedure may have relaxed the patients, reducing anxiety-related biomarkers such as frontal asymmetry. These ambiguities further underline the importance of proper standardisation of biomarker methodologies and validation in large, multisite studies, as emphasised in Saby et al.’s recent review [20].

It is important to note that we restricted our primary analyses in the present study to differences in overall power spectra, though we report the results of exploratory analyses of frequency bands in the Appendix A. Although our findings suggest an impact of IGF1 on electrophysiology, they also acutely underscore the need for standardisation of electrophysiological characterisation techniques, including recording conditions and analytical methods, as well as the need for explicit hypothesis-driven analyses to avoid misleading or apparently contradictory findings.

### 4.3. Network Variability and State-Dependency

We found unexpected differences in network measures between treated and untreated groups at baseline, despite their clinical and genetic homogeneity. These differences raise the possibility of subpopulations of RTT that are differentiated by measures of network function, but not by traditional phenotyping according to clinical and genetic parameters. The precise implications of this observation are yet to be determined. It is possible that subpopulations defined by specific underlying abnormalities at the network level, though driven by a common genetic mutation, will differ in terms of management and prognosis. The greater level of variability at the network level than at the clinical level may have implications for patient stratification and targeted interventions. Interestingly, the regions identified as driving baseline network variability, as well as those predictive of treatment response, were long-range and occipital network connections. We have previously reported on the role of occipital networks in distinguishing classic from variant clinical presentations of RTT [23], which is consistent with preclinical work indicating a primary role for MeCP2 in functional connectivity in the occipital lobe [51] and visual dysfunction in RTT models [52,53]. These networks may be particularly susceptible to network dysfunction secondary to MeCP2 deficit, and may play a role in delineating disease subtype and treatment response and prognosis.

Measurement of occipital network state also accurately predicts treatment outcome, supporting existing literature on visual evoked potential as a predictor of cortical processing deficiencies in RTT [54]. On the other hand, changes in response to IGF1 administration indicate that these cortical networks are plastic and receptive to interventions targeting synaptic activity. These results further support the hypothesis that IGF1 effects are related, at least in part, to network-level changes, and that these effects are dependent on pretreatment network state. This state-dependency of treatment effect highlights the potential of network evaluations for better characterisations of disorder subpopulations for targeted treatments (i.e., biomarker of response). Moreover, while limited by a small sample size due to the rarity of this condition, and the associated risk of overfitting, the high accuracy of the machine learning model for predicting treatment response underscores its potential as a methodology for response biomarkers in IGF1 and other trials in RTT. The identification of specific network measures that could identify drug responders prior to treatment raises the possibility of “precision medicine” approaches for clinical trials for RTT and other neurologic disorders based on network profiles.

### 4.4. Limitations

The present findings are, however, limited by the small number of treated patients, particularly given the unanticipated network heterogeneity at baseline, and by the fact that untreated patients were not randomised or placebo-controlled. Thus, replication of these findings in a larger matched sample would be compelling. Nonetheless, it is worth pointing out that the difference in brain activity was not predicted by the clinical assessment, supporting the need to use additional evaluation criteria for diagnoses. It would be interesting to repeat the IGF1 administration in the same cohorts of treated patients, in particular in the responders, to evaluate the additional changes in network properties induced by repeated treatment.

While differences in network measures involved subtle parameters measuring network interactions (i.e., coherence), it is possible that differences in coarser evaluation metrics such as spectral power would be apparent in larger sample sizes [35]. Moreover, the absence of a formal placebo-controlled randomised study design is a limiting factor. The use of the ISS scale as both a baseline clinical severity measure and an efficacy endpoint, even when applied by independent evaluators, is a limitation when considering that instruments with stronger psychometric properties are currently available [7]. Ultimately, this study highlights the need for greater standardisation of both clinical outcome measures and biomarkers.

## 5. Conclusions

We provide preliminary evidence that IGF1 treatment in RTT is associated with network alterations, and that network profile changes are related to clinical response to treatment. Furthermore, we show that treatment response is state-dependent, based on the pretreatment state of occipital networks. We also demonstrate that characterisation of the state of occipital networks allows accurate prediction to be made of treatment response with a new set of simulated data. These results highlight the potential of network-based assessments for the development of precision medicine paradigms in RTT and other neurologic disorders, and underline the need to incorporate these techniques into larger clinical studies to investigate the potential of network profiling as a biomarker of response to treatment or surrogate endpoint.

## Figures and Tables

**Figure 1 brainsci-10-00515-f001:**
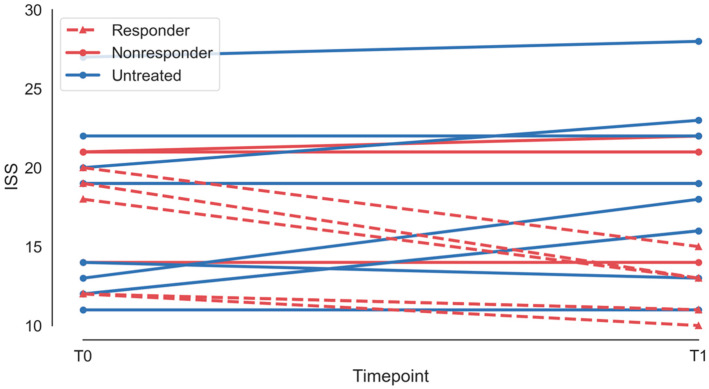
International severity score for treated (red) and untreated (blue) patients at baseline (T0) and at 12 months (T1). Responders are shown with a dashed line, nonresponders with a solid line.

**Figure 2 brainsci-10-00515-f002:**
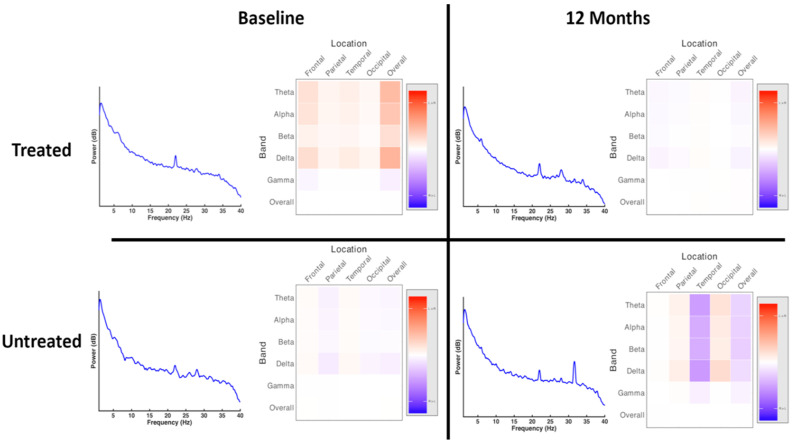
Power spectra and distribution of spectral power at baseline and at 12 months, treated and untreated groups. There were no statistically significant differences in the overall power spectrum or in asymmetry between groups at baseline or at 12 months.

**Figure 3 brainsci-10-00515-f003:**
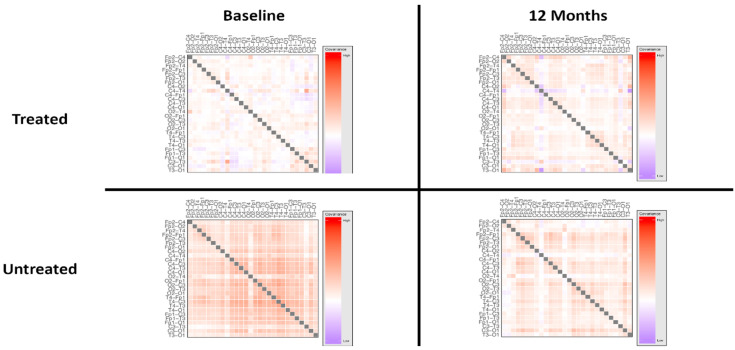
Schematic of network architecture at baseline and at twelve months, treated vs. untreated groups. ANOVA of network loadings (principal components of network measures) demonstrated statistically significant interaction between treatment status and time. There was no statistically significant effect of time alone. Detailed visualisations of network measures are shown in Appendix A.

**Figure 4 brainsci-10-00515-f004:**
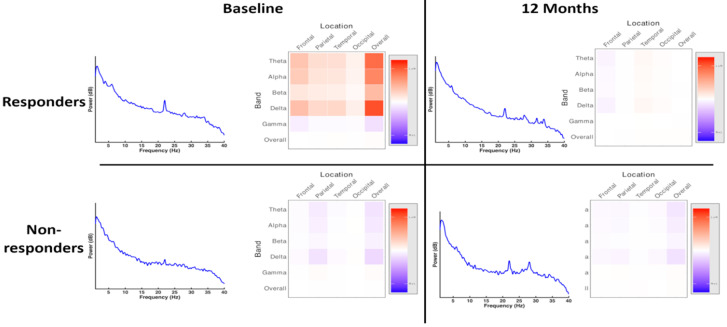
Power spectra and distribution of spectral power at baseline and at 12 months, responders and nonresponders. There were no statistically significant differences in the overall power spectrum between groups at baseline or at 12 months. There were no statistically significant differences in asymmetry measures between or within groups.

**Figure 5 brainsci-10-00515-f005:**
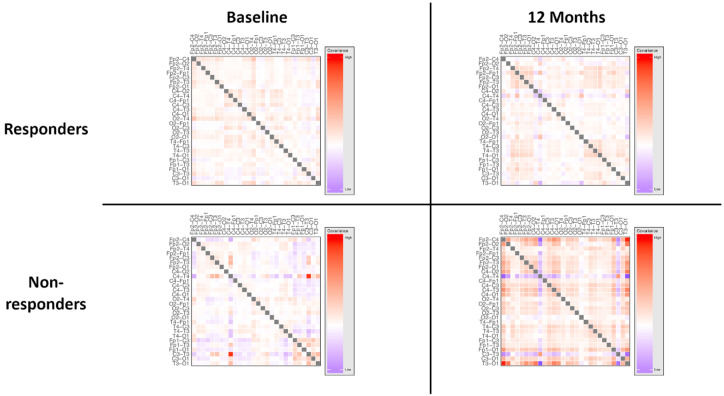
Schematic of network architecture at baseline and at twelve months, responders vs. nonresponders. ANOVA of network loadings (principal components of network measures) demonstrated a statistically significant interaction between treatment status and time, indicating an alteration in network measures following treatment in those that responded clinically, and suggesting a link between electrophysiology and clinical effect. There was no statistically significant effect of time alone. Interestingly, there was evidence of an effect of responder independent of time, suggesting baseline differences in groups that were not apparent on clinical assessment. This may allow EEG biomarkers to be developed which are able to identify patients who are likely to respond to treatment. Detailed visualisations of network measures are shown in Appendix A.

**Figure 6 brainsci-10-00515-f006:**
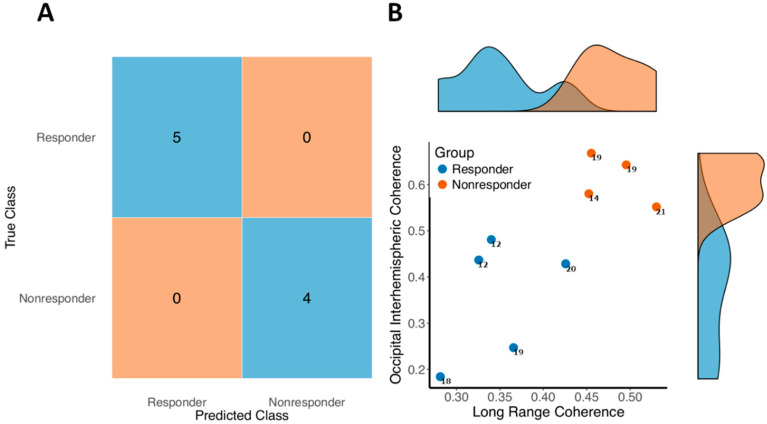
Predictive model performance. A support vector machine trained on computational biomarkers of treatment responsiveness correctly classified novel patient data using a five-fold cross validation approach with 100% accuracy (sensitivity 100%, specificity 100%; *n* = 9) (**A**). The distribution of the predictive biomarkers used for model prediction are shown in (**B**), along with the distributions for each individual predictive variable. The numbers alongside each datapoint show that subject’s ISS score at baseline.

**Table 1 brainsci-10-00515-t001:** Demographic features.

Patient ID	Treatment	Age	Follow-Up	ISS, T1	ISS, T2	Responder
1	Treated	10	14	12	11	Responder
2	Treated	4	13	18	13	Responder
3	Treated	3	11	21	21	Nonresponder
4	Treated	4	7	21	22	Nonresponder
5	Treated	3	12	12	10	Responder
6	Treated	10	7	19	13	Responder
7	Treated	9	15	19	19	Nonresponder
8	Treated	4	13	14	14	Nonresponder
9	Treated	6	14	20	15	Responder
10	Untreated	2	4	12	16	N/A
11	Untreated	2	11	13	18	N/A
12	Untreated	8	13	20	19	N/A
13	Untreated	14	16	14	13	N/A
14	Untreated	6	18	19	19	N/A
15	Untreated	10	12	27	28	N/A
16	Untreated	12	12	20	23	N/A
17	Untreated	12	12	22	22	N/A
18	Untreated	5	12	11	11	N/A

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
