# Peer review of "Functional Network Mapping Reveals State-Dependent Response to IGF1 Treatment in Rett Syndrome"

_brainsci, 2020, doi:10.3390/brainsci10080515_

Round 1
Reviewer 1 Report
In the manuscript entitled “Connectomic profiling reveals state-dependent response to IGF-1 treatment in Rett Syndrome”, the authors aimed at assessing whether IGF1 administration impacts cortical electrophysiology and how network characteristics may predict clinical response to treatment. Even if the study lies on a limited number of patients, the manuscript provides preliminary but important insights on IGF1-mediated therapeutic effects. Moreover, this study indicates the potential of network measure as tool for the stratification affected individuals and as biomarker for clinical trials.
The manuscript's aims are well defined although the narrative should be improved for clarity. For instances, in several cases, figure legends unconventionally include substantial discussion of the data presented: this should be moved elsewhere.
Main points: because of the interesting result indicating unexpected differences in network measures between groups at baseline, the authors should include a correlative analysis between network data and type of mutations shown by enrolled subjects. This could implement the possibility to highlight the biological basis underlying network differences.
Given that the starting point for the EEG analyses and network studies lies on the ability of IGF1 to exert an improvement at a phenotypic level, it could be useful to convert table 1 of the supplementary into a graph placed in the main manuscript showing the severity score changes following IGF1 treatment.
Also, to further support the link between cortical electrophysiology changes and responsiveness to IGF1 administration, the inclusion of a correlation analysis between the ratio of network changes and the amount of responsiveness to treatment could clarify the message of the MS.
The main text is sometime repetitive thus making the understanding of the manuscript difficult: please revise the text in order to make it easier for the reader to find key findings. Also, the combination of the description of 2-way analyses with that of the post-hoc tests, providing an overall explanation of finding, should be implemented.
Furthermore, figure legends are lengthy and, in most cases, instead of describing the figure content, they are a repetition of the main text. Please edit legends accordingly.
Author Response
Response to Reviewer 1
We thank the reviewer for understanding the relevance of our work. Below we address each of the concerns raised. We believe that the revised manuscript has improved compared to the first version.
In the manuscript entitled “Connectomic profiling reveals state-dependent response to IGF-1 treatment in Rett Syndrome”, the authors aimed at assessing whether IGF1 administration impacts cortical electrophysiology and how network characteristics may predict clinical response to treatment. Even if the study lies on a limited number of patients, the manuscript provides preliminary but important insights on IGF1-mediated therapeutic effects. Moreover, this study indicates the potential of network measure as tool for the stratification affected individuals and as biomarker for clinical trials.
The manuscript's aims are well defined although the narrative should be improved for clarity. For instances, in several cases, figure legends unconventionally include substantial discussion of the data presented: this should be moved elsewhere.
The overall structure has been adjusted for clarity. Specifically, the results section has been substantially restructured to provide a clearer picture of the overall results. The discussion elements have been removed from the figure legends.
Main points: because of the interesting result indicating unexpected differences in network measures between groups at baseline, the authors should include a correlative analysis between network data and type of mutations shown by enrolled subjects. This could implement the possibility to highlight the biological basis underlying network differences.
The question of the link between specific mutations and network features is an interesting one. The present group was restricted to patients with MeCP2 mutations (the specific gene mutated has been linked to distinct network features in our previous work- Keogh et al., 2018). Within this small group, there are not sufficient numbers to assess whether there is an association between specific mutations affecting MECP2 mutations and network features – indeed, only two of the patients included show the same mutation, so we can not see clustering based on nature of mutation. If we classify the “severity” of mutation according to Cuddapah et al. 2014, we also see no association (p = 0.55). It is hard to draw conclusions re. the association between specific mutations and network features in this group. This is an important question, but one that requires a larger population to investigate.
Given that the starting point for the EEG analyses and network studies lies on the ability of IGF1 to exert an improvement at a phenotypic level, it could be useful to convert table 1 of the supplementary into a graph placed in the main manuscript showing the severity score changes following IGF1 treatment.
Table 1 has been converted to an addition figure in the main manuscript. This shows ISS before and after treatment for treated and untreated patients, with responders highlighted.
Also, to further support the link between cortical electrophysiology changes and responsiveness to IGF1 administration, the inclusion of a correlation analysis between the ratio of network changes and the amount of responsiveness to treatment could clarify the message of the MS.
Linking the “extent” of network change to magnitude of treatment response is a very interesting question. There is currently no agreed-upon standard way to do this. We have looked at the difference between principal components of pairwise connectivity measures before and after treatment (i.e. difference per subject along main axis of the latent space representing network features) vs. the extent of response to treatment. This showed no correlation in all patients that received treatment (p = 0.84, Pearson’s r), nor in responders alone (p = 0.31). A larger group characterisation may, however, show that these kinds of measures are correlated with treatment response and this was not evident in the small number we had here; further, there is no guarantee that the relationship between network effects and treatment response is linear. Network measures appear to have promise as biomarkers, but characterising their exact relationship to clinical parameters in large groups will be necessary to inform their use.
The main text is sometime repetitive thus making the understanding of the manuscript difficult: please revise the text in order to make it easier for the reader to find key findings. Also, the combination of the description of 2-way analyses with that of the post-hoc tests, providing an overall explanation of finding, should be implemented.
The results section has been substantially rearranged to address this.
Furthermore, figure legends are lengthy and, in most cases, instead of describing the figure content, they are a repetition of the main text. Please edit legends accordingly.
Figure legends have been shortened and discussion elements removed.

Reviewer 2 Report
The study of Keogh et al. is an important clinical preliminary study, suggesting the baseline network variability of Rett Syndrome patients and underlying the importance of individual-based therapies. The force and novelty of the present study is especially the capability to use machine learning to predict the individual response to IGF-1 treatment on the basis of EEG baseline and to underline the importance of therapies designed on the patient background. There are anyway three principal criticisms about this paper:
- Clinical trials with the use of IGF-1 in patients have contradictory results and this issue is not discussed here.
- Results by Keogh et al., are not put in relation to the previous literature background regard EEG recordings in Rett patients. In this sense, it’s important that results by Keogh et al. are compared with the clear perspective done in a recent review by Saby et al., 2020 published on May,28th, where EEG is recognized as a necessary biomarker to predict treatment efficacy according the physiological background of the disease. For example, a previous study of Roche et al., 2019, has measured EEG power spectral differences in a large cohort of RS girls (57), showing a reduction of EEG power in middle frequency bands, and in postregression subjects an increased power in lower frequency bands delta and theta. Moreover, in the paper relative to the phase 2 clinical study done by O'Leary et al., 2018 in a large cohort of patients at Boston Children Hospital, they found EEG worsening at frontal delta, beta and gamma power after 20- week treatment, while Keogh and co-authors report only they ‘did not show clear benefit in the primary outcome measures’ but ‘significant improvements in secondary endpoints’.
- The reduced cohort of patients examined represents a limit of this work. In the previous paper by the same authors (Pini et al., 2016), it is reported that ‘this study should be considered a presentation of clinical cases rather than a clinical trial. Several statistical tests fail to reach significance because of the limited sample size and the statistic test used; hence the replication of study results by additional clinical studies and trials is particularly important.’ I think this comment put by the authors is still valid and especially important in this case to distinguish between responder and not-responder patients, even if I understand the difficulty to recruit patients in one single center in Italy and to follow them within one year from first observation.
Major points:
- This paper follows a paper of Pini et al. 2016 about the effect of mecasermin (Recombinant Human IGF-1) on 10 treated versus 10 untreated RS patients with an improvement both in International Scoring System (ISS) and in Rett Severity Score (RSS) and an enhanced endurance to social and cognitive testing. Is the analysis done in the same group of individuals? In this case why the number is different? Authors should in any case show EEG analysis for a group of tipically developing girls in the same age range for comparison.
- Did the authors explore EEG soon after the end of the treatment (t.i, after 20-24 weeks), so in the middle of the intervening period and if not, why? An intermediate point soon after the end of the treatment would be interesting to clarify the progression of the disease and the ‘acute’ effect of the treatment. Please, comment this point.
- The clinical trial with IG1-1 regarded the use of IGF-1 for 20 weeks (O’Leary et al., 2018). Why did you decided to use a variable time (20-24 weeks)? Is there a difference between the duration of the treatment and the different outcome? This thing should be explained both in the Methods section and in the Discussion.
- Power spectrum in RTT treated and untreated patients at baseline is different according the color map in Power distribution graph on the right at different bands even if the overall spectrum is not different. Why does it happen? It would be interesting to see a graph showing the difference between T1(12 months) and T0 (0 months) in treated and untreated patients both for figure 1 and 2.
- Interhemispheric asymmetry should be discussed better inside the text since now it is only in Supplementary Material.
Minor points:
1)About the title, I think the title of the paper even if cool, it’s misleading, since ‘connectomic profiling’ refers simultaneously to the structural and functional changes at the level of brain mapping network, while this study refers only to functional connectivity, because it is based on EEG recordings and on machine learning to interpret the perspective outcome of IGF-1 treatment on Rett Syndrome patients. Please, correct it with Mapping brain network or something else.
2)At line 65 you should cite also a paper among the others where IGF-1 was shown to be effective in the modulation of structural plasticity in a mouse model of Rett Syndrome (Landi et al., 2011).
3)It is not explained how the patients’ recruitment respect to IGF-1 treatment was done; it seems that the early beginning of the treatment does not correlate with its best outcome, since 3 of 4 non-responders are aged between 3 and 4 years. On the opposite, higher is the ISS, more resistant are patients to treatment (4 of 6). Please, comment this observation within the discussion.
4) Collateral effects of IGF-1 are not discussed and should be described referring to the cohort of patients examined.
5) in Methods section it is not specified how control for EEG noises is kept into consideration in this study (particularly movement of the scalp and eye muscles).
6) Please, express age of Rett patients as median and IQR since it’s not a normal distribution.
7) In figure 5, it would be useful to insert over every point representing a patient the respective value of her ISS index.
8) Since the subdivision between responders and not-responders to IGF-1 treatment is done according ISS index as presented in suppl.fig.1, this info should be added to the Methods section 2.5.1.
Author Response
Response to Reviewer 2
We thank the reviewer for understanding the relevance of our work. Below we address each of the concerns raised. We believe that the revised manuscript has improved compared to the first version.
The study of Keogh et al. is an important clinical preliminary study, suggesting the baseline network variability of Rett Syndrome patients and underlying the importance of individual-based therapies. The force and novelty of the present study is especially the capability to use machine learning to predict the individual response to IGF-1 treatment on the basis of EEG baseline and to underline the importance of therapies designed on the patient background. There are anyway three principal criticisms about this paper:
- Clinical trials with the use of IGF-1 in patients have contradictory results and this issue is not discussed here.
A discussion of the conflicting results of IGF-1 trials has been added, beginning at line 72.
- Results by Keogh et al., are not put in relation to the previous literature background regard EEG recordings in Rett patients. In this sense, it’s important that results by Keogh et al. are compared with the clear perspective done in a recent review by Saby et al., 2020 published on May,28th, where EEG is recognized as a necessary biomarker to predict treatment efficacy according the physiological background of the disease. For example, a previous study of Roche et al., 2019, has measured EEG power spectral differences in a large cohort of RS girls (57), showing a reduction of EEG power in middle frequency bands, and in postregression subjects an increased power in lower frequency bands delta and theta. Moreover, in the paper relative to the phase 2 clinical study done by O'Leary et al., 2018 in a large cohort of patients at Boston Children Hospital, they found EEG worsening at frontal delta, beta and gamma power after 20- week treatment, while Keogh and co-authors report only they ‘did not show clear benefit in the primary outcome measures’ but ‘significant improvements in secondary endpoints’.
A discussion of the existing literature regarding EEG measures in RTT has been added, as has further discussion of the EEG results shown by the studies mentioned. Section beginning line 55 addresses existence of evidence re. EEG measures. Section from lines 540 – 597 discusses results in light of results of previous studies. While the studies appear contradictory, it is possible that the metrics seen in larger studies are not evident in smaller groups such as that used here; further, differences in data acquisition procedure may introduce differences. Our results appear to suggest a role for measures based on functional connectivity, though we emphasise here the importance of standardising methods are characterising these measures in larger populations to resolve the apparently differing results between trials.
- The reduced cohort of patients examined represents a limit of this work. In the previous paper by the same authors (Pini et al., 2016), it is reported that ‘this study should be considered a presentation of clinical cases rather than a clinical trial. Several statistical tests fail to reach significance because of the limited sample size and the statistic test used; hence the replication of study results by additional clinical studies and trials is particularly important.’ I think this comment put by the authors is still valid and especially important in this case to distinguish between responder and not-responder patients, even if I understand the difficulty to recruit patients in one single center in Italy and to follow them within one year from first observation.
Major points:
- This paper follows a paper of Pini et al. 2016 about the effect of mecasermin (Recombinant Human IGF-1) on 10 treated versus 10 untreated RS patients with an improvement both in International Scoring System (ISS) and in Rett Severity Score (RSS) and an enhanced endurance to social and cognitive testing. Is the analysis done in the same group of individuals? In this case why the number is different? Authors should in any case show EEG analysis for a group of tipically developing girls in the same age range for comparison.
This is the same group of individuals. One treated patient (and their control) was not considered for the present analysis as their baseline EEG data was not usable for analysis, so they could not be compared to the rest of the group using electrophysiology. We have updated the text to clarify this point (lines 100-105). Re. the use of a group of typically developing controls: greater characterisation of the nature of network measures in RTT relative to controls is an important question. However, we aim here to answer the question of whether there is a link between electrophysiology and response to IGF-1; unfortunately, we do not have the data currently to explore the nature of network dynamics in RTT relative to controls.
- Did the authors explore EEG soon after the end of the treatment (t.i, after 20-24 weeks), so in the middle of the intervening period and if not, why? An intermediate point soon after the end of the treatment would be interesting to clarify the progression of the disease and the ‘acute’ effect of the treatment. Please, comment this point.
Not all patients underwent full EEG recordings immediately following treatment, so data is not available to compare all patients immediately following treatment. The timepoint used is the first full assessment that all patients underwent. Some underwent additional recording sessions in the interim, but it is difficult to compare these meaningfully across time points.
- The clinical trial with IG1-1 regarded the use of IGF-1 for 20 weeks (O’Leary et al., 2018). Why did you decided to use a variable time (20-24 weeks)? Is there a difference between the duration of the treatment and the different outcome? This thing should be explained both in the Methods section and in the Discussion.
We now clarify the duration of the treatment in the methods section (lines 108-110)
- Power spectrum in RTT treated and untreated patients at baseline is different according the color map in Power distribution graph on the right at different bands even if the overall spectrum is not different. Why does it happen? It would be interesting to see a graph showing the difference between T1(12 months) and T0 (0 months) in treated and untreated patients both for figure 1 and 2.
The plots of distribution of activity shown in figure 1 appear to show some small differences between groups – however, these differences are small in magnitude, highly variable between subjects and fail to reach statistical significance in all cases, even without correction for multiple comparisons. This may suggest a difference in distribution of activity, but our present data do not support that statistically. The actual values and results of t-tests for each pairwise comparison (for all electrode locations and all individual EEG bands) are given in supplementary materials as “exploratory” results. Note that we have added a section in the section “Treatment Effects” that addresses this apparent difference. We have further discussed the nature of these results within the context of current evidence on EEG in RTT in the discussion.
With regard to the difference between T0 and T1 for treated and untreated, we are a bit unclear on what is meant. Figures 1 demonstrates power spectra and distribution for treated and untreated at T0 and T1. If a visualisation of the numeric difference (i.e. values at T1 – T0) is what is meant, I would be concerned that this would be misleading: none of these differences are statistically significant, so any values of T1 – T0 will not be statistically significant from zero, and may suggest relationships which are not borne out in the data.
Figure 2 shows the covariance matrix of network measures at T0 and T1 for treated and untreated. With regard to differences between timepoints, the overall patterns are visible in greater detail in the Supplementary Materials; in terms of visualising differences directly, the subsequent Supplementary Figures demonstrate the pairwise coherence measures themselves across comparisons of T0 and T1 for treated and untreated (as well as responders and nonresponders). This allows for visualisation of the changes in network dynamics between groups / timepoints in a more intuitive way. As with the spectral measures, the individual comparisons of connectivity measures at all locations and bands are provided in Supplementary Tables.
- Interhemispheric asymmetry should be discussed better inside the text since now it is only in Supplementary Material.
Interhemispheric asymmetry was only very briefly discussed as none of the comparisons reached statistical significance for any position or EEG band. This may be a result of the low sample size, however. We have added an acknowledgement of the asymmetry results at line 328 and at line 452. Last, we discuss our results compared with the study performed by Khwaja and colleagues, where they find an asymmetry in the prefrontal region. This asymmetry is usually associated to anxiety, and we discuss how the acquisition procedures in different centres may influence patients’ anxiety (lines 462-465).
Minor points:
1)About the title, I think the title of the paper even if cool, it’s misleading, since ‘connectomic profiling’ refers simultaneously to the structural and functional changes at the level of brain mapping network, while this study refers only to functional connectivity, because it is based on EEG recordings and on machine learning to interpret the perspective outcome of IGF-1 treatment on Rett Syndrome patients. Please, correct it with Mapping brain network or something else.
The title has been altered to reflect this.
2)At line 65 you should cite also a paper among the others where IGF-1 was shown to be effective in the modulation of structural plasticity in a mouse model of Rett Syndrome (Landi et al., 2011).
We have now cited this paper.
3)It is not explained how the patients’ recruitment respect to IGF-1 treatment was done; it seems that the early beginning of the treatment does not correlate with its best outcome, since 3 of 4 non-responders are aged between 3 and 4 years. On the opposite, higher is the ISS, more resistant are patients to treatment (4 of 6). Please, comment this observation within the discussion.
This is an interesting observation; we have acknowledged the trend towards resistance depending on existing ISS rather than age at line 220. Unfortunately the size of our present group limits our ability to interrogate this relationship more thoroughly.
4) Collateral effects of IGF-1 are not discussed and should be described referring to the cohort of patients examined.
Collateral effects of IGF1 have been discussed in previous works and are now cited at line 70
5) in Methods section it is not specified how control for EEG noises is kept into consideration in this study (particularly movement of the scalp and eye muscles).
We have addressed this at line 135, and expanded the appendix to describe how artefacts were dealt with.
6) Please, express age of Rett patients as median and IQR since it’s not a normal distribution.
We have addressed this at line 214.
7) In figure 5, it would be useful to insert over every point representing a patient the respective value of her ISS index.
We have added baseline ISS score to figure 5.
8) Since the subdivision between responders and not-responders to IGF-1 treatment is done according ISS index as presented in suppl.fig.1, this info should be added to the Methods section 2.5.1.
We have added demographic features, including ISS score, to the main text (204-214).

Reviewer 3 Report
Identifying biomarkers and more accurate end-point measures in which to evaluate clinical data is critical for Rett syndrome clinical trials. Despite the small numbers (which is often unavoidable in rare disorders), these measurements hold great promise.
Suggestions:
- Lines 209 and 276: the sentences are long and the statistical explanation is complex leading to the opinion that some words are missing. Kindly address.
- Figure legends: remove 'Interesting' and move any opinions to main text.
- No discussion of mutation-specific effects. Suggest including this with regards to a) affects relative to network alterations and b) affects relative to treatment response.
Author Response
Response to Reviewer 3
We thank the reviewer for understanding the relevance of our work. Below we address each of the concerns raised. We believe that the revised manuscript has improved compared to the first version.
Identifying biomarkers and more accurate end-point measures in which to evaluate clinical data is critical for Rett syndrome clinical trials. Despite the small numbers (which is often unavoidable in rare disorders), these measurements hold great promise.
Suggestions:
- Lines 209 and 276: the sentences are long and the statistical explanation is complex leading to the opinion that some words are missing. Kindly address.
The explanations have been clarified.
- Figure legends: remove 'Interesting' and move any opinions to main text.
Discussion elements have been removed from figure legends.
- No discussion of mutation-specific effects. Suggest including this with regards to a) affects relative to network alterations and b) affects relative to treatment response.
The question of the link between mutation-specific effects on network measures and on treatment response is an interesting one. The present group was restricted to patients with MeCP2 mutations (the specific gene mutated has been linked to distinct network features in our previous work). Within this small group, there are not sufficient numbers to assess whether there is an association between specific mutations affecting MeCP2 and network features – indeed, only two of the patients included show the same mutation, so we can not see clustering based on nature of mutation. If we classify the “severity” of mutation according to Cuddapah et al. 2014, we also see no association with network changes (p = 0.55) or clinical response (p = 0.89). It is hard to draw conclusions re. the association between specific mutations and network features in this group. This is an important question, but one that requires a larger population to investigate.

Round 2
Reviewer 2 Report
I appreciated the answers to my questions and to the questions of the other reviewers and I now think the paper is suitable for publication with minor revision in the text and extensive control of typos.
- Since you analyzed EEG from the same cohort of patients in Pini et al., 2016, please correct at line 111 the number of patients with different treatments. Here, it still appears 10 patients (6, after 20-24 weeks treatment and 4, after 20 weeks treatment) while the subjects analysed are 9. Indeed, in the cover letter, you explain that one treated patient and one control RTT were excluded because their EEG were not analyzable.
- Line 214: median age of 6 years. Please, add ‘years’ and in general revised the text for few typos.
- In the discussion, please add a comment about the possibility to extend/repeat the treatment with IGF-I in the same cohort of patients during their development, especially in responder subjects.
Author Response
Response to reviewer_minor requests
We thank the reviewer and we address the minor points below.
Minor requests:
I appreciated the answers to my questions and to the questions of the other reviewers and I now think the paper is suitable for publication with minor revision in the text and extensive control of typos.
- Since you analyzed EEG from the same cohort of patients in Pini et al., 2016, please correct at line 111 the number of patients with different treatments. Here, it still appears 10 patients (6, after 20-24 weeks treatment and 4, after 20 weeks treatment) while the subjects analysed are 9. Indeed, in the cover letter, you explain that one treated patient and one control RTT were excluded because their EEG were not analyzable.
We now clarify that the patients treated with IGF1 and used in this study are nine lines 112-113.
- Line 214: median age of 6 years. Please, add ‘years’ and in general revised the text for few typos.
We added “years”, and we revised the manuscript for typos.
- In the discussion, please add a comment about the possibility to extend/repeat the treatment with IGF-I in the same cohort of patients during their development, especially in responder subjects.
We added the comment in lines 518-520.
